# A Cost–Consequence Analysis of Nutritional Interventions Used in Hospital Settings for Older Adults with or at Risk of Malnutrition

**DOI:** 10.3390/healthcare12101041

**Published:** 2024-05-17

**Authors:** Alvin Wong, Yingxiao Huang, Merrilyn D. Banks, P. Marcin Sowa, Judy D. Bauer

**Affiliations:** 1Department of Dietetics, Changi General Hospital, Singapore 529889, Singapore; 2School of Human Movement and Nutrition Sciences, University of Queensland, St. Lucia, QLD 4072, Australia; 3Department of Nutrition and Dietetics, Royal Brisbane and Women’s Hospital, Herston, QLD 4029, Australia; 4Centre for the Business and Economics of Health, University of Queensland, St Lucia, QLD 4067, Australia; 5Department of Nutrition, Dietetics and Food, Monash University, Notting Hill, VIC 3168, Australia; judy.bauer@monash.edu

**Keywords:** economic evaluation, nutritional therapy, malnourished, older adults, acute-care settings, complex interventions, DALYs

## Abstract

Background: Malnutrition is a significant and prevalent issue in hospital settings, associated with increased morbidity and mortality, longer hospital stays, higher readmission rates, and greater healthcare costs. Despite the potential impact of nutritional interventions on patient outcomes, there is a paucity of research focusing on their economic evaluation in the hospital setting. This study aims to fill this gap by conducting a cost–consequence analysis (CCA) of nutritional interventions targeting malnutrition in the hospital setting. Methods: We performed a CCA using data from recent systematic reviews and meta-analyses, focusing on older adult patients with or at risk of malnutrition in the hospital setting. The analysis included outcomes such as 30-day, 6-month, and 12-month mortality; 30-day and 6-month readmissions; hospital complications; length of stay; and disability-adjusted life years (DALYs). Sensitivity analyses were conducted to evaluate the impact of varying success rates in treating malnutrition and the proportions of malnourished patients seen by dietitians in SingHealth institutions. Results: The CCA indicated that 28.15 DALYs were averted across three SingHealth institutions due to the successful treatment or prevention of malnutrition by dietitians from 1 April 2021 to 31 March 2022, for an estimated 45,000 patients. The sensitivity analyses showed that the total DALYs averted ranged from 21.98 (53% success rate) to 40.03 (100% of malnourished patients seen by dietitians). The cost of implementing a complex nutritional intervention was USD 218.72 (USD 104.59, USD 478.40) per patient during hospitalization, with additional costs of USD 814.27 (USD 397.69, USD 1212.74) when the intervention was extended for three months post-discharge and USD 638.77 (USD 602.05, USD 1185.90) for concurrent therapy or exercise interventions. Conclusion: Nutritional interventions targeting malnutrition in hospital settings can have significant clinical and economic benefits. The CCA provides valuable insights into the costs and outcomes associated with these interventions, helping healthcare providers and policymakers to make informed decisions on resource allocation and intervention prioritization.

## 1. Introduction

Malnutrition is a significant and prevalent issue in hospital settings, affecting a substantial proportion of hospitalized patients worldwide. It is associated with increased morbidity and mortality, longer hospital stays, higher readmission rates, and greater healthcare costs [1,2,3]. Recent data from the Global Burden of Disease Study 2019 reported age-standardized malnutrition-related disability-adjusted life years (DALYs) at 131 per 100,000 population in 2019 for high-sociodemographic-index (SDI) countries [4]. DALYs combine the years of life lost due to premature death (YLLs) with the years of healthy life lost due to disability (YLDs). The DALY metric has been used to capture a comprehensive picture of the disease burden from malnutrition, which often includes increased mortality risks and chronic health impairments (such as stunted growth, weakened immunity, and developmental delays) [4].

Although high-SDI countries such as Singapore experienced reductions in the age-standardized DALY rates and age-standardized death rate (ASDR) related to malnutrition from 2000 to 2019 [4], it is still important to focus on malnutrition for several reasons. Firstly, malnutrition remains a significant health concern that can lead to a range of negative health outcomes, including increased morbidity, mortality, and healthcare costs. These findings suggest that while high-income nations have lower rates of malnutrition-related DALYs and mortality compared to low-income nations, there is still room for improvement in addressing this issue.

Secondly, this study [4] used a broad definition of malnutrition based on the International Classification of Diseases (ICD) codes for nutritional deficiencies, which may not have captured all forms of malnutrition, such as micronutrient deficiencies or undernutrition due to inadequate dietary intake. It is known that malnutrition coding has not been performed rigorously in hospitals worldwide, which is a possible reason that it has been challenging to establish a true proportion of malnutrition prevalence [5,6,7].

The recent synthesis of primary data from high-quality studies with a low or some concerns regarding the risk of bias indicated that nutritional intervention reduces mortality at the time points of 30 days (30 d), 6 months (6 m), and 12 months (12 m) (relative risk (RR) at 30 d: 0.72, 95%CI: 0.55–0.94; 6 m: 0.81, 95%CI: 0.71–0.92; and 12 m: 0.80, 95%CI: 0.67–0.95) with low heterogeneity (degree of variation observed between the results of individual studies being analyzed) and moderate to high certainty of evidence [8]. Additionally, 33 different variations (education, oral nutritional supplements, or food fortification) and combinations (e.g., education with food fortification or education with exercise) of interventions across 19 reviews were observed [9], indicating that there are many components in an intervention. Even seemingly simple terms such as protected mealtimes (the avoidance of procedures/interventions at patient mealtimes) [10] have widely varied components (e.g., the types of human resources needed, timing, and services provided during protected timing) [8].

Considering the complexity of the interventions, a more comprehensive approach to their development and evaluation for future studies is necessary to determine the optimal combination of resources and strategies to achieve the best clinical outcomes. Besides the complexity of interventions, other factors that may affect the implementation of successful nutritional interventions in hospital settings for populations with or at risk of malnutrition include adherence and the length of time of the interventions.

However, with healthcare budgets under constant pressure, there is a growing need to assess the economic implications of these interventions to ensure the efficient use of resources and maximize patient benefits. The early identification and appropriate management of malnutrition are vital to improving patient outcomes and optimizing the allocation of healthcare resources [11,12]. Health economics tools, such as cost–consequence analysis (CCA), can provide valuable insights into the costs and outcomes associated with different nutritional interventions, allowing healthcare providers and policymakers to make informed decisions regarding resource allocation and intervention prioritization [13]. CCA presents outcomes that are disaggregated, enabling decision-makers to weigh the costs and benefits according to their priorities [13].

Despite the recognized importance of addressing malnutrition in hospitals and the potential impact of nutritional interventions on patient outcomes, there is little research focusing on the economic evaluation of these interventions in the hospital setting. This study aims to fill this gap by (1) conducting a cost–consequence analysis to compare the costs and outcomes of nutritional interventions targeting malnutrition in the hospital setting, using data from recent systematic reviews and meta-analyses, and (2) providing evidence-based recommendations for healthcare providers and policymakers on the efficient allocation of resources for nutritional interventions in hospital settings, considering both costs and patient outcomes.

## 2. Methods

This study adheres to the Consolidated Health Economic Evaluation Reporting Standards (CHEERS) 2022 guidance, which provides a comprehensive framework for the reporting of health economic evaluations. The CHEERS 2022 checklist was used to ensure that all relevant aspects of the study design and findings were reported transparently and comprehensively [14].

### 2.1. Study Design and Population

This study employed a cost–consequence analysis (CCA) to evaluate the economic implications of various nutritional interventions targeting malnutrition in hospital settings, compared to the standard (usual care in hospital with normal meals or placebo). The analysis focused on older adults with or at risk of malnutrition in the hospital setting. We excluded patient populations requiring highly specialized care (such as critically ill, oncology, and palliative care patients), those from developing countries, and those unable to consume nutrients orally and primarily receiving parenteral (PN) and enteral nutrition (EN) support, as these interventions are life-sustaining and may have systematically different outcomes. A CCA table was constructed to display each intervention’s costs and outcome measures for ease of comparison.

The study’s perspective was that of Singapore’s healthcare system, considering both the direct costs and consequences associated with the provision of nutritional interventions and the indirect costs related to these patient outcomes, such as hospital readmissions and lengths of stay. The time horizon extended to 12 months post-discharge, which allowed for the assessment of both the immediate and longer-term impacts of the interventions.

### 2.2. Data Sources and Extraction

#### 2.2.1. Clinical Outcomes (Consequences)

A comprehensive literature search was conducted to identify systematic reviews and meta-analyses providing data on the effectiveness and costs of the nutritional interventions under investigation based on a recent umbrella review and meta-analysis [8]. Primary studies that used validated malnutrition screening and assessment tools were used in the meta-analysis of clinical outcomes. The base case scenario considered complex nutritional interventions versus standard care, as presented in the umbrella review, for the outcomes of 30-day (30 d), 6-month (6 m), and 12-month (12 m) mortality; 30 d and 6 m hospital readmission; hospital complications; and lengths of stay [8].

The outcome measures were expressed in their original units, with DALYs calculated based on the age-standardized malnutrition-related DALYs at 131 per 100,000 population in 2019 for high-sociodemographic-index (SDI) countries including Singapore. Information on the patient load seen within the Singapore Health Services (SingHealth), an academic medicine cluster of four tertiary hospitals, five national specialty centers, three community hospitals, and nine community polyclinics, covering 50% of Singapore’s patient population, was used in the CCA [15]. The three main institutions of Singapore General Hospital, Changi General Hospital, and Sengkang General Hospital have a total of 169,019 patient admissions. We excluded the data from KK Women’s and Children’s Hospital as this analysis excluded the pediatric population [15]. Data were extracted into an Excel worksheet (Version 16.84, Microsoft, Redmond, WA, USA) by one author (AW) and checked for accuracy by a second author (HY).

#### 2.2.2. Statistical Analysis for Clinical Outcomes

Post hoc subgroup analyses were performed using a recently published methodology for the characterization of the complexity of nutritional interventions [9]. The intervention strategies were grouped under the three main areas of education and training (ET), exogenous nutrient supply (ENS), and environment and services (ES) [9]. The framework and multivariate regression model were applied on primary RCTs identified in the umbrella review [8], and comparing complex (ENS, ET, and ES) versus simple nutritional intervention (ENS or ET only) to determine the pooled effect size for 6-month mortality. We excluded ES-only interventions as the umbrella review did not identify any interventions that incorporated ES only. Meta-analytic methods were reported in the published umbrella review [8]. The meta-analysis was performed based on the recommendations from the Cochrane Handbook for Systematic Reviews of Interventions [16] and using the statistics software Review Manager Version 5.4.1 (Copenhagen: The Nordic Cochrane Centre, The Cochrane Collaboration, 2014).

#### 2.2.3. Cost Analysis

The cost components for each intervention were standardized to a common currency (USD, median with interquartile range (IQR)) and base year (2023) using appropriate inflation rates and currency conversion factors. The costs were divided into two categories: (1) costs incurred during the hospital stay and (2) costs associated with continuing the intervention upon discharge for up to six (6) months. The resources required for nutritional interventions were identified from the recently published study characterizing the complexity of nutritional interventions [9], and the list of resources is reported in the Appendix A. In summary, the resources included human resources for the performance of the intervention or provision of training, nutritional intervention products such as special hospital meals and food for special medical purposes (e.g., oral nutritional and food fortification supplements) [17,18], equipment/tools/software (e.g., anthropometry measurement tools, specialized biochemical analyses, dietary analysis programs, exercise and therapy needs), and other miscellaneous or overhead costs, such as transportation costs, parking fees, and training materials. Additionally, costs incurred from exercise and therapy interventions that were provided as part of the intervention were presented separately. All calculations and cost analyses were performed using Excel (Microsoft, USA). This division allowed for a more detailed analysis of the costs associated with each intervention and helped to identify areas where cost savings could be achieved.

#### 2.2.4. Sensitivity Analysis

Sensitivity analyses were performed to assess the robustness of the findings and the impact of uncertainty in the input parameters, including cost and clinical outcome estimates. For clinical outcomes, sensitivity analyses were conducted by varying the following parameters: (a) the treatment success rate reported in the literature [19,20] and (b) the proportions of malnourished patients seen by dietitians based on local hospital data. For the sensitivity analyses of costs, the parameters for the type of human resources and intensity/frequency of intervention visits (allied health professionals, registered/enrolled nurses, general practitioners/specialists, ancillary staff, and volunteers), type of facilities and services used for post-discharge interventions (outpatient clinic, telehealth services, and home visitations), equipment used (anthropometry measurement, laboratory/biochemical tests, training, therapy, and exercise), and length of intervention [8] were considered. The sensitivity analyses for cost outcomes were performed using Excel (Microsoft, USA) and for clinical outcomes with Review Manager Version 5.4.1 (Copenhagen: The Nordic Cochrane Centre, The Cochrane Collaboration, 2014).

## 3. Results

The results of the cost–consequence analysis for each nutritional intervention are presented in Table 1. The table displays the standardized costs (USD, IQR, Year 2023) and clinical outcome measures for each intervention.

### 3.1. Cost Outcomes

The cost of implementing a complex nutritional intervention encompassing ENS, ET, and ES (base case scenario) is USD 218.72 (IQR: USD 4.59, USD 478.40) per patient during hospitalization (median length of stay = 5 days), with an additional cost of USD 814.27 (IQR: USD 397.69, USD 1212.74) when the intervention is extended to the community (home visits), USD 487.90 (IQR: USD 234.51, USD 723.19) for outpatient settings, and USD 431.62 (IQR: USD 206.37, USD 638.77) if followed up with telehealth/telemedicine.

Additionally, if exercise or therapy is included as part of the nutritional intervention, the cost of a complex intervention during hospitalization increases to USD 299.23 (IQR: USD 218.72, USD 739.08). This increases to USD 638.77 (USD 602.05, USD 1185.90) if the exercise or therapy component is continued in the community setting due to the increase in human resources needed to carry out the intervention, but at a lower cost of USD 432.62 (USD 404.79, USD 862.07) if telehealth is used as the medium for intervention.

The per patient cost of an intervention also depends on the type of healthcare worker involved in implementing the intervention, with the highest initial cost required for registered nurses at USD 4009.16 (IQR: USD 1943.61, USD 8018.31), followed by healthcare assistants or enrolled nurses at USD 2208.13 (IQR: USD 1043.09, USD 4416.26), and USD 718.30 (IQR: USD 292.67, USD 1425.58) if the intervention is carried out by volunteers. The sensitivity analyses indicated that the cost of the intervention would be lower with simple nutritional interventions of ET (USD 81.14, IQR: USD 44.63, USD 117.65) or ENS (USD 91.69 IQR: USD 49.64, USD 131.95) only. The clinical benefits associated with nutrition diminished to insignificant differences over standard care for ET- and ENS-only interventions.

### 3.2. Consequences (Clinical Outcomes)

The clinical outcomes from the original umbrella review [8] are summarized in Table 1 and Appendix A. Nutritional interventions reduced mortality at 30 d (15 studies, *n* = 4156, RR: 0.72, 95%CI: 0.55–0.94, *p* = 0.02, low heterogeneity, and high certainty of evidence), 6 m (27 studies, *n* = 6387, RR: 0.81, 95%CI: 0.71–0.92, *p* = 0.001, low heterogeneity, and moderate certainty of evidence), and 12 m (27 studies, *n* = 6387, RR: 0.80, 95%CI: 0.67–0.95, *p* = 0.01, low heterogeneity, and moderate certainty of evidence). However, the additional subgroup analysis (Appendix A) performed for the CCA on the length of the intervention indicated that nutritional interventions lasting more than three (3) months were more likely to reduce overall mortality when compared to shorter interventions.

No significant reduction (RR: 0.83 (0.67–1.02)) in 6 m readmissions for interventions was reported in the original meta-analysis [8]. Interventions >3 m may lead to a 26% reduction in 6 m readmissions (RR: 0.74 (0.55–1.01)) [8]. There was no significant improvement or difference in ENS-only interventions, and there was insufficient evidence for the effects of ET-only interventions due to the limited clinical studies available. There was insufficient information related to the effect of therapy/exercise interventions when used in nutritional trials.

The CCA indicated that 28.15 disability-adjusted life years (DALYs) were averted across the Singapore Health Services’ (SingHealth, Singapore) institutions of Singapore General Hospital, Changi General Hospital, and Seng Kang General Hospital due to the successful treatment or prevention of malnutrition by dietitians from 1 April 2021 to 31 March 2022. This was based on an estimated 45,000 patients (from existing data, 27% of patients were on oral nutrition with malnutrition) [21,22], with 68% (range: 53% to 83%) of the malnourished patients returning to a well-nourished state after treatment [19,20].

The sensitivity analyses to evaluate the impact of varying success rates in treating malnutrition (53% and 83%) and the proportions of malnourished patients seen by dietitians (60%, 80%, and 100%) indicated that the total DALYs averted ranged from 21.98 (53% success rate) to 40.03 (100% of malnourished patients seen by dietitians) (Table 1).

**Table 1 healthcare-12-01041-t001:** Cost–consequence analysis of nutritional interventions in hospital settings for adult patients with or at risk of malnutrition.

Standardized Costs [USD (IQR), Base Year 2023]					
Inpatient Nutritional Intervention					
Factors	Details	Cost of Nutritional Intervention per Patient During Hospitalization			Assumptions and Sources
Intervention Costs in Hospital Per Patient	Median Length of Intervention = 5 days	Base Case Scenario: Complex Intervention (ENS|ES|ET) USD 218.72 (USD 104.59, USD 478.40) Sensitivity Analysis: With Exercise/Therapy in Ward USD 299.23 (IQR: USD 218U.72, SD 739.08)	ET Only USD 81.14 (USD 44.63, USD 117.65)	ENS Only USD 91.69 (USD 49.64, USD 131.95)	Based on median length of stay of 5 days (from umbrella review [8], primary studies, and SingHealth 2021/2022 data) [15] Exercise/therapy session conducted by a physiotherapist, occupational therapist, or exercise physiologist.
**Initial Start-Up Costs for Nutritional Intervention**					
Nutritional Equipment and Programs (Pre-Implementation)	Per Ward	Base Case Scenario: Advanced Software and Equipment USD 3358.16 (USD 1679.08, USD 5037.24) Sensitivity Analysis: Basic Software or Equipment USD 2365.41 (USD 1182.70, USD 3548.11)	ET Only Nil Costs Involved	ENS Only Nil Costs Involved	Assuming that one set of equipment is required for each ward of 40 patients in a hospital.
Staff Training Program (Pre-Implementation, Frequency Variable)	Per Year	Base Case Scenario: Intervention by Registered Nurse USD 4009.16 (USD 1943.61, USD 8018.31) Sensitivity Analysis: Healthcare Asst/Enrolled Nurse USD 2208.13 (USD 1043.09, USD 4416.26) Sensitivity Analysis: Volunteers USD 718.30 (USD 292.67, USD 1425.58)	ET Only Nil Costs Involved	ENS Only Nil Costs Involved	Assuming that training is required once a year.
**Outpatient Nutritional Intervention**					
Intervention Costs Post-Discharge Per Patient	Median Length of Intervention = 3 months	Base Case Scenario: Followed Up with Home Visits USD 814.27 (USD 397.69, USD 1212.74) Sensitivity Analysis: With Exercise/Therapy at Home USUSD 638.77 (USD 602.05, USD 1185.90) Sensitivity Analysis: Followed Up at Outpatient Clinic USD 487.90 (USD 234.51, USD 723.19) Sensitivity Analysis: Followed Up with Telehealth USD 431.62 (USD 206.37 USD 638.77)	ET Only NA	ENS Only USD 671.91 (USD 323.62, USD 993.82)	(1) Clinic space already available and set aside for outpatient follow-up. (2) Telehealth services in place post-COVID-19 implementation in all SingHealth institutions. (3) Home visits involved dietitian and nurse visits, using a taxi as the main transportation, and visiting at different periods. (4) Exercise/therapy session conducted by a physiotherapist, occupational therapist, or exercise physiologist.
**Consequences (Clinical Outcomes)**					
Mortality	30 day	Intervention Period: During Hospitalization Only 28% Reduction (RR: 0.72 (0.55–0.94))	ET Only Insufficient Information	ENS Only No Difference	Main and post hoc analysis of umbrella review [8].
	6 month	Intervention Period: Any 19% Reduction (RR: 0.81 (0.71–0.92)) Intervention Periods of At Least 3 Months 27% Reduction (RR: 0.73 (0.60–0.88))	ET Only Insufficient Information	ENS Only No Difference	Main and post hoc analysis of umbrella review [8].
	12 month	Intervention Period: Any 20% Reduction (RR: 0.80 (0.67–0.95)) Intervention Periods of At Least 3 Months 27% Reduction (RR: 0.73 (0.60–0.90))	ET Only Insufficient Information	ENS Only No Difference	Main and post hoc analysis of umbrella review [8].
Readmissions	30 day	Intervention Period: Any No Difference	ET Only Insufficient Information	ENS Only Insufficient Information	Main and post hoc analysis of umbrella review [8].
	6 month	Intervention Period: Any 17% Reduction (RR: 0.83 (0.67–1.02)) Intervention Periods of At Least 3 Months 26% Reduction (RR: 0.74 (0.55–1.00))	ET Only Insufficient Information	ENS Only 21% Reduction	Main and post hoc analysis of umbrella review [8].
Hospitalization Related	Complications	Intervention Period: Any No Difference	ET Only Insufficient Information	ENS Only No Difference	Main and post hoc analysis of umbrella review [8].
	Length of Stay	Intervention Period: Any No Difference	ET Only Insufficient Information	ENS Only No Difference	Main and post hoc analysis of umbrella review [8].
Nutritional Related	Nutritional Status	ENS|ES|ET Mean: 68% (range: 53% to 83%) of patients progressed from malnourished to well nourished [19,20] Mean: 68% (range: 59.8% to 76.7%) of patients progressed from at risk to no risk of malnutrition [23,24,25]	ET Only 76% of patients progressed from malnourished to well nourished in one study only [19]	ENS Only 25% of patients progressed from malnourished to well nourished in one study only [26]	Post hoc analysis of umbrella review [8].
DALYs	Potential DALYs Averted	ENS|ES|ET 28.15 DALYs averted across the SingHealth institutions due to the successful treatment of malnutrition by dietitians from 1 April 2021 to 31 March 2022 Sensitivity Analysis: Total DALYs averted for each scenario: (a) 53% treatment success rate—21.98 DALYs averted; (b) 83% treatment success rate—34.22 DALYs averted; (c) 60% of malnourished patients seen—24.01 DALYs averted; (d) 80% of malnourished patients seen—32.02 DALYs averted; (e) 100% of malnourished patients seen—40.03 DALYs averted.	ET Only Insufficient Information	ENS Only Insufficient Information	(1) DALYs per 100,000 patients for malnutrition in Singapore: 131; (2) Percentage of patients in Singapore hospitals who are malnourished: 35% (27% on oral nutrition, 8% on enteral nutrition) [22]; (3) Percentage of malnourished patients seen by dietitians: 70% (personal data, Changi General Hospital Internal Audit); (4) Success rate in eliminating malnutrition: 68% [19,20]; (5) 166,671 patients were seen across tertiary institutions of Changi General Hospital, Seng Kang General Hospital, and Singapore General Hospital [15].

Asst: assistant; DALYs: disability-adjusted life years; ET: education and training; ENS: exogenous nutrient supply; ES: environment and services; NA: not applicable.

## 4. Discussion

The cost–consequence analysis (CCA) of nutritional interventions in hospital settings highlighted the potential for significant improvements in patient outcomes through the effective treatment and prevention of malnutrition. The cost of implementing a complex nutritional intervention encompassing education and training (ET), exogenous nutrient supply (ENS), and environment and services (ES) is USD 218.72 (USD 104.59, USD 478.40) per patient during hospitalization, with an additional USD 814.27 (USD 397.69, USD 1212.74) per patient when the intervention is extended to community and outpatient settings. If exercise or therapy is included as part of the multi-component intervention over a 3-month period, an additional USD 638.77 (USD 602.05, USD 1185.90) per patient is incurred.

These results indicate that complex nutritional interventions, encompassing ET, EN, and ES, are more expensive than simple interventions that only involve ET or EN. This finding is important because it implies that complex interventions may need more resources but may also lead to more significant clinical benefits. Notably, the clinical benefits associated with nutritional interventions diminish to insignificant differences over standard care for both ET-only and EN-only interventions. This finding underscores the importance of a comprehensive approach to nutritional interventions, which includes multiple components rather than a single aspect. However, as there were insufficient studies available for ET, EN, and ES with concurrent exercise or therapy observed in the umbrella review, we were unable to report outcomes associated with this combined intervention [8].

Simplistic interventions such as providing oral nutritional supplements only, without any follow-up by clinicians or education on dietary modifications, while easy to carry out, may not be effective in the longer term. This is consistent with the recent work from Baldwin et al [27], where the authors reported that the evidence for the effects of ONS in patients with or at risk of malnutrition is uncertain.

In terms of clinical outcomes, implementing a nutritional intervention for patients with or at risk of malnutrition reduces the mortality rates at all time points, with moderate to high certainty of evidence [8]. Additionally, 28.15 disability-adjusted life years (DALYs) were averted across the three SingHealth institutions due to dietitians’ successful management of malnutrition from 1 April 2021 to 31 March 2022. DALYs were used for this CCA instead of the more commonly known quality-adjusted life years (QALYs), which focus more on quality of life and treatment outcomes [13]. Hence, QALYs may not adequately capture the long-term and multifaceted nature of malnutrition’s impact, particularly in vulnerable populations, where the disease burden manifests through a combination of premature mortality and long-term disability [22].

The results from the sensitivity analyses for DALYs also stressed the importance of identifying malnutrition, providing interventions, and ensuring the successful treatment of patients to reduce the burden of disease. Improving patients’ access to nutritional support and interventions through reimbursement has been shown to improve clinical outcomes in a local cohort study of patients receiving medical financial support [21]. Additionally, the complexity of nutritional interventions needs to be recognized to better tailor interventions to the needs of specific patient populations [9]. This finding reinforces that a more comprehensive approach to intervention development and evaluation may be crucial in achieving optimal clinical outcomes, considering the unique needs and preferences of different patient populations.

While the malnutrition-related DALYs and mortality have been decreasing in the past decade and are predicted to decrease further, with obesity superseding it, it is unlikely that disease-related malnutrition will be fully eradicated [4]. Hence, disease-related malnutrition will continue to be an economic burden. Therefore, it is still essential to implement targeted and deliberate interventions that address the underlying causes and contributing factors. By optimizing these factors, healthcare providers can improve patient outcomes and reduce the healthcare burden.

Our results highlight the need and urgency to prevent disease- and diet-related malnutrition through more community-based malnutrition screening and assessments and to target at-risk populations at the early stages. This will align with the Healthier SG initiative [28], launched in 2022, a comprehensive national plan to address the increasing prevalence of chronic diseases, the ageing population, and rising healthcare costs in Singapore. The initiative prioritizes preventive health to boost community wellness. It encompasses engaging doctors, creating tailored health plans, partnering with local entities, starting a broad sign-up drive, and forming vital infrastructure and funding models [28,29]. Furthermore, as the Singapore healthcare funding system moves towards a capitation-based funding model [30,31], healthcare clusters are incentivized to focus on preventive care and regional population health management, and the prudent use of healthcare resources will be a priority [28].

### Limitations

Despite being comprehensive, this study has some limitations. Firstly, the CCA only examined some nutritional interventions and patient groups, not covering all strategies to address malnutrition in hospital settings, such as parenteral and enteral nutrition [32]. There was also insufficient information for nutritional interventions performed in conjunction with therapy or exercise, a common treatment modality in clinical practice but not sufficiently included or reported in nutritional trials. However, as nutritional interventions are complex [13,32], categorizing and classifying interventions into ENS, ET, and ES will reduce the heterogeneity, a strength of our CCA. Future research should focus on targeted, patient-centered interventions considering the diverse patient needs, cultural factors, and resource availability.

Secondly, the heterogeneity of the derived cost data may affect the findings’ generalizability. The standardized costs and outcome measures used for the comparison may not capture the resource use and effectiveness variations across different healthcare contexts and funding mechanisms [9,17]. Future research should use cost data that are relevant to the context and examine variations in cost estimates.

Thirdly, the clinical outcomes were based on systematic reviews and meta-analyses, which may display variations due to the different study designs and patient populations [33,34]. The meta-analyses included studies with varying risks of bias, possibly affecting the true intervention effects’ representation [8]. To mitigate this issue, we included only studies with a low risk of bias or some concerns in the meta-analytic results. Additionally, we could not determine whether long-term nutritional interventions improved the quality of life in patients receiving the intervention, similar to the findings of Soderstrom et al. [35]. Evidence suggests that while nutritional interventions may enhance clinical outcomes, prolonged use may impact quality of life [36,37].

Finally, the analysis did not consider potential healthcare system differences and funding mechanisms [17], implementation strategies, and other contextual factors affecting nutritional interventions’ success [36,37], as we primarily adopted the perspective of Singapore’s healthcare system. Implementing effective nutritional interventions requires interdisciplinary collaboration among healthcare providers, researchers, policymakers, and other stakeholders [38]. This requires knowledge sharing, commitment to evidence-based decision-making, and continuous intervention effectiveness evaluation. Such collaborative efforts can significantly improve hospital malnutrition management, leading to improved patient outcomes and sustainable healthcare.

Future research should consider staff training, patient adherence, and resource availability. It is observed that the integration of health technology and digital health solutions may also enhance nutritional interventions’ cost-effectiveness [39]. This includes telemedicine and remote monitoring technologies, which can allow the more efficient and timely delivery of care [40], and digital tools for patient education and self-management to improve adherence and compliance with nutritional recommendations [38,41,42].

## 5. Conclusions

Our analysis of the costs and consequences demonstrated that treating and preventing malnutrition effectively in hospital settings can significantly improve patient outcomes. The sensitivity analyses emphasize the importance of increasing the number of malnourished patients who receive treatment and the success rates of these treatments. Future research should focus on comparing the cost-effectiveness of different nutritional interventions and patient groups and examining the influence of contextual factors on intervention success. By building on the findings of this study and addressing its limitations, healthcare providers and policymakers can make more informed decisions about how to use resources and which interventions to prioritize, ultimately leading to improved patient outcomes and a reduced burden on the healthcare system.

## Data Availability

The data that support the findings of this study are available in the Appendix A of this article.

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
