# Peer review of "A Cost–Consequence Analysis of Nutritional Interventions Used in Hospital Settings for Older Adults with or at Risk of Malnutrition"

_healthcare, 2024, doi:10.3390/healthcare12101041_

Round 1

Reviewer 1 Report

Comments and Suggestions for Authors

Thank you for extending the invitation to review this manuscript. Entitled "Cost-Consequence Analysis of Nutritional Interventions Used in Hospital Settings for Older Adults with or At Risk of Malnutrition," the authors present a good piece of work. It is evident that considerable effort has been invested in its composition, resulting in a fluid and engaging read. Upon careful consideration, I find the proposed methodology holds significant scientific promise, with the potential to contribute substantially to the discourse in healthcare journal. Nonetheless, few minor concerns warrant attention before the paper can be deemed suitable for publication.

1.     The authors have addressed the Cost Consequence Analysis, which is good in its own. However, I wonder whether this is different that Cost Effectiveness analysis, the commonly used technique in economics evaluation. If so, how?

2.     Besides, the authors have used DALYs as measure for heath outcomes. I wonder why, given QALYs is the most common health outcome in the area of health Economics. Please advise. Also, the reader would appreciate a brief definition for DALYs?

3.     In “Consequences (Clinical Outcomes)” section, the authors have used I^2. What is that and how is calculated and why?

4.     The paper lacks the implication of the methods/results for future use of these methods and modelling work in this area. Please elaborate.

Author Response

Authors’ Reply to Reviewer 1

 Thank you for taking the time to review our manuscript, A Cost-Consequence Analysis of Nutritional Interventions Used in Hospital Settings for Older Adults with or At Risk of Malnutrition. We appreciate your thoughtful and constructive comments, which have been invaluable in enhancing the quality of our work. Below, we have provided detailed responses to each of your comments.

Reviewer’s Comment

  1. The authors have addressed the Cost Consequence Analysis, which is good in its own. However, I wonder whether this is different that Cost Effectiveness analysis, the commonly used technique in economics evaluation. If so, how?

Author’s Reply

As indicated in the introduction section, cost-consequence analysis (CCA), can provide valuable insights into the costs and outcomes associated with different nutritional interventions, allowing healthcare providers and policymakers to make informed decisions regarding resource allocation and intervention prioritization.(13) CCA presents outcomes disaggregated, enabling decision-makers to weigh the costs and benefits according to their priorities, versus the more commonly known Cost-Effectiveness Analysis which typically reports cost per QALY, which is dependent on the quality of life, that has not shown to improve in nutritional trials despite improvement in other clinical outcomes.

Reviewer’s comment

  1. Besides, the authors have used DALYs as measure for health outcomes. I wonder why, given QALYs is the most common health outcome in the area of health Economics. Please advise. Also, the reader would appreciate a brief definition for DALYs?

Author’s Reply

We have added the following definition of DALYs to the introduction as well as an explanation on why DALY is preferred over QALY.

“DALYs combine the years of life lost due to premature death (YLLs) with the years of healthy life lost due to disability (YLDs). The DALY metric has been used to capture a comprehensive picture of the disease burden from malnutrition, which often includes increased mortality risk and chronic health impairments (such as stunted growth, weakened immunity, and developmental delays). (4)”

Reviewer’s comments

  1. In “Consequences (Clinical Outcomes)” section, the authors have used I^2. What is that and how is calculated and why?

Author’s Reply

The I2 is a measure of heterogeneity used in meta-analysis and was originally presented in the pooled effects (Relative Risk) in the manuscript. We have realised that this may cause confusion to the readers and hence removed the I2 statistic  from the entire revised manuscript and replaced with the following explanation in the introduction

“nutritional intervention reduces mortality ……. with a low heterogeneity (degree of variation observed between the results of individual studies being analyzed)”

Reviewer’s comments

  1. The paper lacks the implication of the methods/results for future use of these methods and modelling work in this area. Please elaborate.

Author’s Reply

We have significantly increased the description of the methodology to allow replication of the calculations, and added further discussion on how the results relate to Singapore’s change of healthcare financing model as well as the national healthcare roadmap to manage chronic diseases.

Once again, thank you for your thorough review and valuable suggestions. We believe the manuscript has been significantly improved thanks to your insights, and we hope the revised version meets your expectations.

Sincerely,

Alvin Wong

Dietetics Department, Changi General Hospital, Singapore

School of Human Movement and Nutrition Sciences, The University of Queensland, Australia

Reviewer 2 Report

Comments and Suggestions for Authors

Dear authors, 

The manuscript is interesting, but I have some major issues to address prior to publication. See my comments below.

- Please, reformat the draft using only 3rd person language.

- There are several abbreviations throughout the entire manuscript that impair the readability. Please, reduce those to the strictly necessary.

- Review the ponctuation of the entire text. Some typos were detected.

- Did you perform the sensitivity analysis using the Excel software? Please, clarify.

- The sensitivity analysis was not clear enough in the methods section. Please, clarify.

- The table is larger than usual. Please, consider insert the data in a supplementary file. Restrict your results to the most important data.

- Clarify the following sentence: "the clinical benefits associated with these inter ventions were not significantly different from standard care, suggesting that the cost-ef fectiveness of these interventions may be limited." How the cost-effectiveness is limited as the cost is low, with the same benefit?

- The discussion is too short. I strongly recommend the authors to compare the current findings in a deeply manner to the literature.

Author Response

Authors’ Reply to Reviewer 2

Thank you for taking the time to review our manuscript, A Cost-Consequence Analysis of Nutritional Interventions Used in Hospital Settings for Older Adults with or At Risk of Malnutrition. We appreciate your comments, which have been invaluable in enhancing the quality of our work, and incorporated the recommendations into the revised manuscript. Below, we have provided detailed responses to each of your comments.

Reviewer’s Comments

  1. The manuscript is interesting, but I have some major issues to address prior to publication. See my comments below.

- Please, reformat the draft using only 3rd person language.

Authors’ Reply

We have reformatted the draft in 3rd person where possible.

Reviewer’s Comments

  1. There are several abbreviations throughout the entire manuscript that impair the readability. Please, reduce those to the strictly necessary.

Authors’ Reply

We have reduced the abbreviations where possible  .

Reviewer’s Comments

  1. Review the punctuation of the entire text. Some typos were detected.

Authors’ Reply

We have corrected the punctuation errors where possible, however, some errors will require post-editing by the editorial office.

Reviewer’s Comments

  1. Did you perform the sensitivity analysis using the Excel software? Please, clarify.

- The sensitivity analysis was not clear enough in the methods section. Please, clarify

Authors’ Reply

We have added the following to the Methods section

“Sensitivity analyses were performed to assess the robustness of findings and the impact of uncertainty in input parameters, including cost and clinical outcome estimates. For clinical outcomes, sensitivity analyses were conducted by varying the following parameters of (a) treatment success rate reported in the literature (19, 20) and (b) proportions of malnourished patients seen by dietitians based on local hospital data. For sensitivity analyses of costs, the following parameters for the type of human resources and intensity/frequency of intervention visits (allied health professionals, registered/ enrolled nurses, general practitioners/specialists, ancillary staff and volunteers), type of facilities and services used for post-discharge interventions (outpatient clinic, telehealth services, and home visitations), equipment used (anthropometry measurement, laboratory / biochemical tests, training, therapy and exercise), and length of intervention (8) were considered. Sensitivity analyses for cost outcomes were performed using Excel (Microsoft, USA) and clinical outcomes with Review Manager Version 5.4.1 (Copenhagen: The Nordic Cochrane Centre, The Cochrane Collaboration, 2014).”

Reviewer’s Comments

  1. The table is larger than usual. Please, consider insert the data in a supplementary file. Restrict your results to the most important data.

Authors’ Reply

We have moved Table 1 to supplementary material and have adjusted the size of Table 2. Further resizing can be done by the editorial office post-editorial.

Reviewer’s Comments

  1. Clarify the following sentence: "the clinical benefits associated with these interventions were not significantly different from standard care, suggesting that the cost-effectiveness of these interventions may be limited." How the cost-effectiveness is limited as the cost is low, with the same benefit?

Authors’ Reply

We have rewritten this to:

“These results indicate that complex nutritional interventions, encompassing ET, EN, and ES, are more expensive than simple interventions that only involve ET or EN. This finding is important because it implies that complex interventions may need more resources but may also lead to more significant clinical benefits. Notably, the clinical benefits associated with nutritional interventions diminished to insignificant differences over standard care for both ET-only and EN-only interventions. This finding underscores the importance of a comprehensive approach to nutritional interventions, which includes multiple components rather than a single aspect.”

Reviewer’s Comments

  1. The discussion is too short. I strongly recommend the authors to compare the current findings in a deeply manner to the literature

Authors’ Reply

We have significantly increase the content of the discussion, firstly to incorporate how the CCA results relate to the Healthier SG initiative on chronic diseases management as well as the recent change in healthcare funding mechanisms in Singapore.

Please refer to the revised discussion section in the manuscript.

Once again, thank you for your thorough review and valuable suggestions. We believe the manuscript has been significantly improved thanks to your insights, and we hope the revised version meets your expectations.

Sincerely,

Alvin Wong

Dietetics Department, Changi General Hospital, Singapore

School of Human Movement and Nutrition Sciences, The University of Queensland, Australia

Reviewer 3 Report

Comments and Suggestions for Authors

Review comments and suggestions for authors

Section 1: Introduction

The importance of malnutrition in hospital environments and the possible advantages of nutritional interventions are emphasized in the introduction. It offers a thorough synopsis of the problem, including pertinent research and worldwide statistics. The issue gains legitimacy when current results from the Global Burden of Disease Study 2019 are included. Nonetheless,

      For correctness, the phrase "resyntheses" should be changed to "syntheses". Furthermore, the findings would be more clearly expressed if the measures of heterogeneity in meta-analyses were better elaborated upon. For correctness, the word "combinations" should be used instead of "permutations".

      To give a more precise description, the sentence "Even seemingly simple terms such as Protected Mealtimes have widely varied components" may be defined for readers who may not be familiar with the term

      The discussion on the complexity of interventions and the importance of economic evaluation is well-articulated. However, there could be more explanation provided regarding the specific components of interventions, such as "Protected Mealtimes," to aid understanding for readers less familiar with the terminology.

Section 2: Methodology section

The demographic, data sources, extraction techniques, and study design are all covered in detail in the methodology section. The study's transparency and credibility are improved by conforming to the Consolidated Health Economic Evaluation Reporting Standards (CHEERS) 2022 recommendations.

      To shed light on the study population, it would be better to define "at risk of malnutrition" and its corresponding criteria in the statement "The analysis focused on adult patients with or at risk of malnutrition in the hospital setting."

      It would be more accurate to define "interventions are life-sustaining" to mean that certain patient populations—such as those who are unable to consume nutrients orally—need parenteral (PN) and enteral nutrition (EN) support.

      To improve comprehension, it would be beneficial to define "standardized costs" in the statement "A CCA table was constructed to display the standardized costs and outcome measures for each intervention."

      To prevent confusion, the statement "The study's perspective is that of the healthcare provider" should make it clear if it relates to a particular healthcare system or kind of provider.

      For the sentence "The time horizon extends to 6 months post-discharge," it would be prudent to justify the selection of this particular time horizon and its significance to the goals of the study.

      To prevent misunderstanding, it's crucial to make it clear if the phrase "The overall analysis for nutritional interventions considered them as one overarching intervention" refers to a conceptual or statistical grouping.

      The presentation of the data table disrupts the formatting of the manuscript due to its size. Please review the proportions and the rows and columns of the table.

Section 3 : Results section

      To avoid confusion, it would be more correct to state whether the beginning cost mentioned in the sentence "Registered nurses require the highest initial cost at US$4,009.16" is per nurse or per patient.

      To better inform readers, define "standard care" in the statement "However, the clinical benefits associated with these interventions were not significantly different from standard care."

      To provide clarity in the sentence "Some of the clinical outcomes have been described in our earlier work for the umbrella review," the term "umbrella review" needs to be defined right away.

      It would be better to make it explicit in the sentence "In summary (Table 2 and Figure 1), nutritional intervention reduced mortality at 30d" what is included in the summary and how it connects.

      The phrase "interventions of any length of time" might be more precisely defined in the sentence "No significant reduction [RR: 0.83 (0.67-1.02)] in 6 months readmissions for interventions of any length of time was observed" in order to avoid confusion.

      It would be more lucid to identify the source or methodology of the sensitivity analysis in the statement "Sensitivity analysis suggested that intervention periods of at least 3 months may lead to a 26% reduction in readmissions."

Section 4 : Discussion section

      The cost-consequence analysis (CCA) results are succinctly summarized in this part, which also emphasizes the significance of complete dietary therapy in enhancing patient outcomes. Certain points, nevertheless, still need to be clarified and improved.

      Although there are some uncertainties in the presentation, the part offers insightful information about the economic implications and efficacy of dietary therapy. To improve understanding, it is necessary, for example, to define specific terms and acronyms and to provide the sources of contemporary literature that are cited.

      Furthermore, even though the restrictions are suitably addressed, the variability of cost data and its consequences for generalizability may need further explanation. Further elaboration on the aspects taken into account during the sensitivity assessments would enhance the discussion.

      It would be redundant to include the term "for interventions of any length of time" in the statement "No significant reduction [RR: 0.83 (0.67-1.02)] in 6 months readmissions was observed."

      It would be easier to avoid confusion in the sentence "The CCA showed that while complex nutritional interventions encompassing ENS, ET, and ES were associated with higher costs" if the terms "ENS," "ET," and "ES" were spelled out clearly at the beginning.

Section 5 : Conclusion section

The necessity for future research to concentrate on assessing the cost-effectiveness of various nutritional therapies and patient populations is duly noted in this section.

      To more effectively direct future research efforts, it would be beneficial to clarify the precise characteristics of cost-effectiveness that should be assessed.

      Furthermore, although if the section talks about overcoming the study's limitations, it might be reinforced by including more detailed instructions on which constraints need to be addressed and how to do so in future research.

Author Response

Authors’ Reply to Reviewer 3

Thank you for taking the time to review our manuscript, A Cost-Consequence Analysis of Nutritional Interventions Used in Hospital Settings for Older Adults with or At Risk of Malnutrition. We appreciate your comments, which have been invaluable in enhancing the quality of our work, and incorporated the recommendations into the revised manuscript. Below, we have provided detailed responses to each of your comments.

Reviewer’s Comments

Introduction

The importance of malnutrition in hospital environments and the possible advantages of nutritional interventions are emphasized in the introduction. It offers a thorough synopsis of the problem, including pertinent research and worldwide statistics. The issue gains legitimacy when current results from the Global Burden of Disease Study 2019 are included. Nonetheless,

For correctness, the phrase "resyntheses" should be changed to "syntheses”. Furthermore, the findings would be more clearly expressed if the measures of heterogeneity in meta-analyses were better elaborated upon. For correctness, the word "combinations “should be used instead of "permutations".

Author’s Reply

Thank you for pointing this out. We have the adjustment made. This has been incorporated into the revised manuscript on page 2

Reviewer’s Comments

To give a more precise description, the sentence "Even seemingly simple terms such as Protected Mealtimes have widely varied components" may be defined for readers who may not be familiar with the term.

Author’s Reply

The definition has been incorporated into the revised manuscript on page 2

Reviewer’s Comments

− The discussion on the complexity of interventions and the importance of economic evaluation is well-articulated. However, there could be more explanation provided regarding the specific components of interventions, such as "Protected Mealtimes," to aid understanding for readers less familiar with the terminology.

Author’s Reply

Please refer to our response above.

Reviewer’s Comments

Methodology

The demographic, data sources, extraction techniques, and study design are all covered in detailing the methodology section. The study's transparency and credibility are improved by conforming to the Consolidated Health Economic Evaluation Reporting Standards (CHEERS) 2022 recommendations. To shed light on the study population, it would be better to define "at risk of malnutrition “and its corresponding criteria in the statement "The analysis focused on adult patients with or at risk of malnutrition in the hospital setting."

Author’s Reply

This has been incorporated into the revised manuscript on page 3.

Reviewer’s Comments

It would be more accurate to define "interventions are life-sustaining" to mean that certain patient populations—such as those who are unable to consume nutrients orally—need parenteral (PN) and enteral nutrition (EN) support.

Author’s Reply

We have amended the sentence to

“those unable to consume nutrients orally and primarily receiving parenteral (PN) and enteral nutrition (EN) support, as these interventions are life-sustaining.”

Reviewer’s Comments

To improve comprehension, it would be beneficial to define "standardized costs" in the statement "A CCA table was constructed to display the standardized costs and outcome measures for each intervention."

Authors’ Reply

We have addressed the above by defining the standardized costs in the Cost Analysis Section in Methods.

Reviewer’s Comments

To prevent confusion, the statement "The study's perspective is that of the healthcare provider" should make it clear if it relates to a particular healthcare system or kind of provider.

Authors’ Reply

We have amended to “The study's perspective is that of Singapore’s healthcare system,”

Reviewer’s Comments

For the sentence "The time horizon extends to 6 months post-discharge," it would be prudent to justify the selection of this particular time horizon and its significance to the goals of the study.

Authors’ Reply

We apologise for the error as it should read “The time horizon extends to 12 months post-discharge, which allows for the assessment of both the immediate and longer-term impacts of the interventions.

Reviewer’s Comments

To prevent misunderstanding, it's crucial to make it clear if the phrase "The overall analysis for nutritional interventions considered them as one overarching intervention" refers to a conceptual or statistical grouping.

Authors’ Reply

We have amended the sentence toThe base case scenario considered complex nutritional interventions versus standard care”

Reviewer’s Comments

The presentation of the data table disrupts the formatting of the manuscript due to its size. Please review the proportions and the rows and columns of the table.

Authors’ Reply

We have attempted to adjust the size of the Table and will work with the editorial office to ensure a suitable table size is achieved post-editorial.

Reviewer’s Comments

Results section

To avoid confusion, it would be more correct to state whether the beginning cost mentioned in the sentence "Registered nurses require the highest initial cost at US$4,009.16"is per nurse or per patient.

Authors’ Reply

We have amended the sentence to “The per patient cost of intervention also depends on the type of healthcare worker involved in implementing the intervention”

Reviewer’s Comments

 To better inform readers, define "standard care" in the statement "However, the clinical benefits associated with these interventions were not significantly different from standard care."

Authors’ Reply

We have incorporated the above recommendations in the manuscript.

”compared to standard (usual care in hospital with normal meals or placebo).”

Reviewer’s Comments

To provide clarity in the sentence "Some of the clinical outcomes have been described in our earlier work for the umbrella review," the term "umbrella review" needs to be defined right away.

Authors’ Reply

We have incorporated the above recommendations in the manuscript.

Reviewer’s Comments

It would be better to make it explicit in the sentence "In summary (Table 2 and Figure 1), nutritional intervention reduced mortality at 30d" what is included in the summary and how it connects.

Authors’ Reply

We have rewritten the sentence to “The clinical outcomes from the original umbrella review (8) is summarized in Table 1 and Figure 1.”

Reviewer’s Comments

 It would be more lucid to identify the source or methodology of the sensitivity analysis in the statement "Sensitivity analysis suggested that intervention periods of at least 3 months may lead to a 26% reduction in readmissions."

Authors’ Reply

We have provided the citations and sources in Table 1 as well as in the results section.

Reviewer’s Comments

Discussion section

The cost-consequence analysis (CCA) results are succinctly summarized in this part, which also emphasizes the significance of complete dietary therapy in enhancing patient outcomes. Certain points, nevertheless, still need to be clarified and improved.

Although there are some uncertainties in the presentation, the part offers insightful information about the economic implications and efficacy of dietary therapy. To improve understanding, it is necessary, for example, to define specific terms and acronyms and to provide the sources of contemporary literature that are cited.

Authors’ Reply

Where possible, we have added definitions, explanations and examples on specific terms and acronyms, as well as citations for the definitions.

Reviewer’s Comments

Furthermore, even though the restrictions are suitably addressed, the variability of cost data and its consequences for generalizability may need further explanation. Further elaboration on the aspects taken into account during the sensitivity assessments would enhance the discussion.

Authors’ Reply

We have further elaborated the methodology for sensitivity analysis as well as indicated in Table 1 what parameters were involved in the sensitivity analysis.

Reviewer’s Comments

It would be redundant to include the term "for interventions of any length of time" in the statement "No significant reduction [RR: 0.83 (0.67-1.02)] in 6 months readmissions was observed."

Authors’ Reply

We have amended the sentence as suggested.

Reviewer’s Comments

The phrase "interventions of any length of time" might be more precisely defined in the sentence "No significant reduction [RR: 0.83 (0.67-1.02)] in 6 months readmissions for interventions of any length of time was observed" in order to avoid confusion.

Authors’ Reply

Refer to above response

Reviewer’s Comments

It would be easier to avoid confusion in the sentence "The CCA showed that while complex nutritional interventions encompassing ENS, ET, and ES were associated with higher costs" if the terms "ENS," "ET," and "ES" were spelled out clearly at the beginning.

Authors’ Reply

We have incorporated it as follows:

“The cost-consequence analysis (CCA) of nutritional interventions in hospital settings highlighted the potential for significant improvement in patient outcomes through the effective treatment and prevention of malnutrition. The cost of implementing a complex nutritional intervention encompassing education and training (ET), exogenous nutrient supply (ENS), and environment and services (ES)”

Reviewer’s Comments

Conclusion

The necessity for future research to concentrate on assessing the cost-effectiveness of various nutritional therapies and patient populations is duly noted in this section.

− To more effectively direct future research efforts, it would be beneficial to clarify the precise characteristics of cost-effectiveness that should be assessed.

− Furthermore, although if the section talks about overcoming the study's limitations, it might be reinforced by including more detailed instructions on which constraints need to be addressed and how to do so in future research.

Authors’ Reply

We have incorporated further discussion on why DALYs should be used as an outcome measure in malnutrition, and significantly rewritten the Limitations section to incorporate the above suggestions.

Once again, thank you for your thorough review and valuable suggestions. We believe the manuscript has been significantly improved thanks to your insights, and we hope the revised version meets your expectations.

Sincerely,

Alvin Wong

Dietetics Department, Changi General Hospital, Singapore

School of Human Movement and Nutrition Sciences, The University of Queensland, Australia

Reviewer 4 Report

Comments and Suggestions for Authors

Dear Authors

It has been a pleasure to have the opportunity to review your manuscript “A Cost-Consequence Analysis of Nutritional Interventions Used in Hospital Settings for Older Adults with or At Risk of Malnutrition”.

Here are some recommendations that may help to improve the manuscript, both in terms of content and compliance with the "Instructions for Authors" required by this journal:

·       In first place, it is advisable to read the manuscript presentation "Instructions for Authors". For example, the citation in ALL text must be placed in square brackets and before the full stop that closes the sentence [1,2].

·       Although the journal is not demanding in the presentation format (as long as it complies with the sections within the manuscript), for peer reviewing, it facilitates the presentation in the template format(Microsoft Word template) allowing the reviewer and author to identify the line number where the recommendations are made.

Introduction. In the following statement "Our recent resyntheses of primary data from high-quality studies with low or some concerns regarding risk of bias indicated that nutritional intervention reduces mortality at time-points of 30 days, ..." cite high-quality studies [x,x,x]. During this paragraph there are inconsistencies, it indicates "studies" but at the end of the paragraph, it only indicates a citation. In this paragraph, you also refer to 19 revisions but do not cite them. In case you have used a revision document that includes 19 revisions, cite that document.

In the following objective “(2) to provide evidence-based recommendations for healthcare providers and policymakers on the efficient allocation of resources for nutritional interventions in hospital settings, considering both costs and patient outcomes.” Why was only review used? A review is based on previous studies, where clinical studies have great relevance. May have missed published research not included in the reviews. Also, ignore recent publications that are not found in reviews.

Methods. 2.1. Study Design and Population . “This study employed a cost-consequence analysis (CCA) to evaluate the economic implications of various nutritional interventions targeting malnutrition in hospital settings …” It indicates that cost-consequence analysis was used, but what type of methodology was used in this work? Narrative/systematic/meta-analysis literature review?

Review the first paragraph included in these sections, “2.2. Data Sources and Extraction 2.2.1. Clinical Outcomes (Consequences)”, it seems to include information missing in the previous section (Study Design). Place in "study design" the type of study and the target population to be studied. In 2.2.1 focus on where and how the data are obtained...

Adapt the format of the tables to the journal's recommendations, see the "Instructions for Authors". Adjust the width of the column to the content, since the current table is not at all visual for publication. Remove the links within the “source of cost” column and replace with an appropriate citation. You can also use the journal's Microsoft Word template.

On page 4 it indicates “General Hospital and Sengkang General Hospital have a total of 169019 patient admissions. We excluded the data from KK Women’s and Children’s Hospital as this analysis excludes the paediatric population.(15) Data were extracted into an Excel worksheet (Microsoft, USA) by one author (AW) and checked for accuracy by a second author (HY)” Does this statement mean that you accessed hospital records/medical records? If you accessed the records, did you have approval from the health center? And if you accessed the medical records, ethics, and data protection committee?

Ethical aspects. If your answer is affirmative in the previous section that refers to access to hospital data, do not forget to include the relevant authorizations here. If the data used is from a previous investigation where the data has been used again, you must detail this aspect and indicate the number of prior positive authorizations and the institution that granted it (Ethics Committee, hospital xxx).  

References. Review all bibliographic references. Again go to the "Instructions for Authors", indicated as recommended by the ACS style guide. E.g:

1. Abizanda P, Sinclair A, Barcons N, Lizán L, Rodríguez-Mañas L. Costs of Malnutrition in Institutionalized and 103 Community-Dwelling Older Adults: A Systematic Review. J Am Med Dir Assoc. 2016;17(1):17-23.

1. Abizanda, P., Sinclair, A., Barcons, N., Lizán, L., Rodríguez-Mañas. L. Costs of Malnutrition in Institutionalized and 103 Community-Dwelling Older Adults: A Systematic Review. J Am Med Dir Assoc. 2016, 17(1), 17-23.

This document has shortcomings, and the authors need to take the time to rework it. 

I wish you good luck with your manuscript.

Greetings

Author Response

Authors' Reply to Reviewer 4

Thank you for your detailed feedback on our manuscript, A Cost-Consequence Analysis of Nutritional Interventions Used in Hospital Settings for Older Adults with or At Risk of Malnutrition. We appreciate the time and effort you have taken in providing your recommendations, and have responded to each of your comments below.

Reviewer’s Comment: Here are some recommendations that may help to improve the manuscript, both in terms of content and compliance with the "Instructions for Authors" required by this journal: In first place, it is advisable to read the manuscript presentation "Instructions for Authors". For example, the citation in ALL text must be placed in square brackets and before the full stop that closes the sentence [1,2]. Although the journal is not demanding in the presentation format (as long as it complies with the sections within the manuscript), for peer reviewing, it facilitates the presentation in the template format (Microsoft Word template) allowing the reviewer and author to identify the line number where the recommendations are made.

Authors' Reply: The citation format in the revised manuscript is aligned with the Healthcare editorial office's guidelines, and we trust they will handle any required formatting edits.

Reviewer’s Comment: Introduction: In the following statement "Our recent resyntheses of primary data from high-quality studies with low or some concerns regarding risk of bias indicated that nutritional intervention reduces mortality at time-points of 30 days, ..." cite high-quality studies [x,x,x]. During this paragraph there are inconsistencies, it indicates "studies" but at the end of the paragraph, it only indicates a citation. In this paragraph, you also refer to 19 revisions but do not cite them. In case you have used a revision document that includes 19 revisions, cite that document.

Authors' Reply: The cited study is an umbrella review and meta-analysis that encompassed 19 papers, and we used the pooled results. Hence, it is unnecessary to individually cite the 19 papers, as the meta-analysis provides the relevant comprehensive data.

Reviewer’s Comment: In the following objective, "(2) to provide evidence-based recommendations for healthcare providers and policymakers on the efficient allocation of resources for nutritional interventions in hospital settings, considering both costs and patient outcomes,” why was only a review used? A review is based on previous studies, where clinical studies have great relevance. May have missed published research not included in the reviews. Also, ignore recent publications that are not found in reviews.

Authors' Reply: Our analysis is based on an umbrella review and meta-analysis that we published earlier. This comprehensive work systematically retrieved, assessed, and synthesized all relevant clinical trials, including recent ones. While some new studies might not have been included, the umbrella review remains an authoritative source, offering an evidence-based foundation for our cost-consequence analysis.

Reviewer’s Comment: Methods. 2.1. Study Design and Population: "This study employed a cost-consequence analysis (CCA) to evaluate the economic implications of various nutritional interventions targeting malnutrition in hospital settings…" It indicates that cost-consequence analysis was used, but what type of methodology was used in this work? Narrative/systematic/meta-analysis literature review? Review the first paragraph included in these sections, "2.2. Data Sources and Extraction 2.2.1. Clinical Outcomes (Consequences)," it seems to include information missing in the previous section (Study Design). Place in "study design" the type of study and the target population to be studied. In 2.2.1 focus on where and how the data are obtained…

Authors' Reply: Please refer to the revised manuscript's methodology section, which clarifies the nature of a cost-consequence analysis. The manuscript also addresses your concerns regarding study design and target population.

Reviewer’s Comment: Adapt the format of the tables to the journal's recommendations, see the "Instructions for Authors." Adjust the width of the column to the content, since the current table is not at all visual for publication. Remove the links within the "source of cost" column and replace with an appropriate citation. You can also use the journal's Microsoft Word template.

Authors' Reply: Refer to our reply to Point 2. The table has been moved to the Supplementary Material and formatted accordingly.

Reviewer’s Comment: On page 4 it indicates "General Hospital and Sengkang General Hospital have a total of 169,019 patient admissions. We excluded the data from KK Women’s and Children’s Hospital as this analysis excludes the pediatric population.(15) Data were extracted into an Excel worksheet (Microsoft, USA) by one author (AW) and checked for accuracy by a second author (HY)." Does this statement mean that you accessed hospital records/medical records? If you accessed the records, did you have approval from the health center? And if you accessed the medical records, ethics, and data protection committee?

Authors' Reply: The citation above refers to the publicly available annual report from the healthcare cluster, and the data extracted into Excel was obtained from non-medical sources and from the meta-analysis. As detailed in both the original and revised manuscripts, this is a cost-consequence analysis based on meta-analysis results. Therefore, no data were derived directly from hospital or medical records, and no ethics approval or data protection review was required.

Reviewer’s Comment: References. Review all bibliographic references. Again go to the "Instructions for Authors," indicated as recommended by the ACS style guide. E.g:

Abizanda P, Sinclair A, Barcons N, Lizán L, Rodríguez-Mañas L. Costs of Malnutrition in Institutionalized and 103 Community-Dwelling Older Adults: A Systematic Review. J Am Med Dir Assoc. 2016;17(1):17-23. Abizanda, P., Sinclair, A., Barcons, N., Lizán, L., Rodríguez-Mañas. L. Costs of Malnutrition in Institutionalized and 103 Community-Dwelling Older Adults: A Systematic Review. J Am Med Dir Assoc. 2016, 17(1), 17-23.

Authors' Reply: Refer to our reply for Point 2. We will work with the editorial office to ensure all references meet the journal's style guide.

Once again, thank you for your review efforts. We look forward to working with the editorial office to make the necessary adjustments.

Sincerely,
A Wong
Changi General Hospital, Singapore
The University of Queensland, Australia

Round 2

Reviewer 2 Report

Comments and Suggestions for Authors

Dear authors, 

Thank you for all the improvements in your current version.

Regards.

Author Response

Authors' reply to Reviewer: We thank you for the time and peer review efforts to improve this manuscript. 

Reviewer 4 Report

Comments and Suggestions for Authors

Dear Authors

You have worked hard to improve the documents. You have provided an answer to all the suggestions I made in the previous review. In addition, a more visually clean presentation is much appreciated.

Remind them of the need to review the citation and reference standards (see Chicago and MDPI styles). In both styles, the text is cited in square brackets [x], and the references, check the italics in the names of the journals and the year in bold.

Wish you the best of luck with your publication.

Greetings

Author Response

Authors' reply to Reviewer: We thank you for the time and peer review efforts to improve this manuscript. We will work with the editorial office to ensure that the formatting is consistent.